# Fulminant Recurrent Thrombosis in a Patient with Catastrophic Antiphospholipid Syndrome and Its Thirty-Day Outcome

**Pierpaolo Di Micco** [1,*], **Maurizio Dorato** [1], **Maurizio Latte** [2], **Maria D'Antò** [1], **Vittorio Luiso** [1] and **Gerolamo Sibilio** [1]

1    AFO Medica, UOC Medicina, P.O. Santa Maria delle Grazie, ASL Napoli 2nord, 80076 Pozzuoli, Italy; maurizio.dorato@aslnapoli2nord.it (M.D.); mariadantp@libero.it (M.D.); vittorio.luiso@gmail.com (V.L.); girolamo.siblio@gmail.com (G.S.)

2    Unit of Nephrology, UOC Medicina, P.O. Santa Maria delle Grazie, ASL Napoli 2nord, 80076 Pozzuoli, Italy; maurizio.latte@aslnapoli2nord.it

\*    Correspondence: pdimicco@libero.it

**Abstract:** Catastrophic antiphospholipid syndrome (CAPS) is a rare clinical form of antiphospholipid syndrome (APS) associated with life-threatening complications due to simultaneous thrombosis that may affect small and large vessels. It may be localized to the venous and/or arteries at the same time, and there are not available guidelines based on randomized clinical trials or large series. We here report a clinical case of CAPS with onset after resolution of oligo-symptomatic infection SARS-CoV-2, that had transient improvement with warfarin after recurrent thromboses occurred despite treatment off-label with low doses of low molecular weight heparin. Furthermore, we tried to trace a line by which a multidisciplinary team may set specific timing to have follow-up because of the high morbidity, mortality, and prolonged time of hospitalization.

**Keywords:** antiphospholipid syndrome; catastrophic antiphospholipid syndrome; recurrent thrombosis; venous thromboembolism; thrombotic microangiopathies

## 1. Background

Catastrophic antiphospholipid syndrome (CAPS) is considered a rare clinical presentation of antiphospholipid syndrome (APS) associated with life-threatening complications due to simultaneous thrombosis that may affect small and large vessels [1]. It may be localized to the venous and/or arteries at the same time [2]. There are not available guidelines based on randomized clinical trials, while consensus on diagnosis and treatments is based on clinical registries or large series [3]. From a clinical point of view, CAPS may also hit the vasa of organs, inducing frequently specific organ failure [4].

Epidemiologically, according to data available in the EURO Phospholipid project [5] and CAPS registry [3], only 1% of patients with APS develop CAPS. Yet, the incidence of CAPS in specific cohorts of patients with autoimmune disease is nearly 2%, and it also includes CAPS without anamnestic APS [1,6]. Patients with CAPS, in fact, show recent contacts with surgery, infection, or newly diagnosed cancer independently of the presence of APS [7].

In the case we report together to an insidious onset of disease with multiple thrombotic events and severe thrombocytopenia, we also had a needful support from the nephrological unit because haemodialysis has been a fundamental therapeutic support. Yet, we pointed out a series of clinical steps to follow-up on the reported complex clinical scenario because, as in other cases available in the literature, the long-term follow-up of CAPS may be hard to find among the involved different specialities.

## 2. Case History

In the August 2023, D.E., a 76 year-old woman affected by chronic renal insufficiency with home care treatment based on atorvastatin 20 mg, darbopoietin 4000 three times

weekly, and Ramipril 5 mg daily, felt a sudden pain localized to the right flank and leg. Nearly fifteen days before, she recovered after a SARS-CoV-2 infection with respiratory symptoms but without lung failure; she was treated with the mRNA anti-SARS-CoV-2 vaccine 2 years before with 1 regular dose and two booster doses. She took oral NSAIDs to manage lower limb pain without clinical benefit, and after 10 h from the onset, she was referred to the emergency department and admitted to the emergency room. After the first clinical evaluation with a negative Blumberg sign and a positive Wells score (3 points), the patient was asked to perform an abdominal CT scan that revealed renal vein thrombosis and proximal deep venous thrombosis (DVT) of the right leg associated with severe thrombocytopenia, anaemia, mild prolongation of activated partial thromboplastine time (aPTT), and renal failure with a strong increase in BUN (blood samples are summarized in Table 1). She was referred to haemodialysis after the placement of a central venous catheter in the left femoral vein, and because of the severe thrombocytopenia, according to guidelines, she did not take any type of antithrombotic treatment. Yet, after a few hours, a new acute pain in the left leg revealed a recurrence of venous thromboembolism for proximal deep venous thrombosis. A new blood sample confirmed severe thrombocytopenia, anaemia and mild prolongation of aPTT. Because of the recurrent VTE with thrombocytopenia and prolonged aPTT, immediately a screening for antiphospholipid antibodies, lupus anticoagulant, and antinuclear antibodies was ruled and tested positive (results summarized in Table 2), and in the meantime, a low dose of enoxaparin was started (i.e., 40 mg daily). Thrombophilic screening is reported in Table 2, and differential diagnosis was performed with thrombotic microangiopathies, also testing levels of ADAMTS-13. Unfortunately, the day after, a new acute pain in the left foot with cyanosis and skin ulceration of two fingers appeared as clinical signs of arterial thrombosis of the distal anterior left tibial artery. For this reason, a diagnosis of CAPS was performed because of the presence of arterial and venous thrombosis that affected the legs and organs (thrombotic events are summarized in Table 3). She was referred to the placement of a central venous line in the right internal jugular vein to perform haemodialysis and pulse therapy with steroids (i.e., methylprednisolone 2000 mg daily for the first 2 days, 1000 mg for the other 2 days, and 500 mg the day after) and intravenous immunoglobulins (IVIG) (i.e., 0.4 g/kg/day × 5 days), but maintained a reduced dose of enoxaparin (i.e., 40 mg twice daily) because of the simultaneous presence of severe thrombocytopenia and kidney failure that are contraindications to antithrombotic treatment with full doses of heparins. Unfortunately, a further VTE recurrence was detected in the right internal jugular vein after the first 24 h, so we decided to stop enoxaparin and start warfarin because, in the meantime, platelets raised over 30,000 for mmcube. In this phase of clinical evolution and decision-making, we tried to perform intensive treatments because CAPS usually shows fast progression with a worse outcome. For this reason, we also considered treatments including plasma exchanges as far as apheresis and ECMO, but unfortunately, they were unavailable in our hospital. Yet, 48 h after the beginning of pulse therapy, platelets raised 57,000 mmcube and warfarin raised a right range of 2.2 (INR range 2.0–3.0), so she placed a new central venous line in the left subclavian vein to perform new haemodialysis. A progressive clinical improvement regarding the trend toward recurrent thrombosis, the grade of thrombocytopenia, and pain control was detected, and fever and dyspnoea were not present for several days. Then, after 5 days, steroids' pulse therapy and IVIG were stopped; in those days, biomarkers of inflammation decreased, although they remained higher than normal. INR values were in the therapeutic range for 1 week, and nonrecurrent thrombosis was detected by the patient. Yet, in order to go on with steroids' treatment, metilprednisolone 1 mg\kg\iv daily was started following 3 weeks and associated rituximab (each 3 weeks at low doses because the reduced GFR, 875 mg iv planned each 4 weeks), as specific monoclonal antibodies against CD20+ lymphocytes were begun. Haemodialysis went on for the following 3 weeks with suboptimal clinical results because blood pressure was unstable. For this reason, the patient, after nearly 3 weeks, had a clinical evolution toward anasarca (e.g., bilateral palpebral oedema associated with lower limb upper limb oedema) until dyspnoea relapsed as an objective clinical sign of heart failure. Heart failure

showed an inexorable trend with a reduced response to haemodialysis, and this trend was confirmed by the daily increase of biomarkers as pro-BNP. Furthermore, inflammation biomarkers rose to their initial values, and platelets decreased to 18,000 mmcube. Despite haemodialysis and medical support for thrombocytopenia and oral anticoagulation on day 24 from the onset of the disease, the clinical scenario shows progressive heart failure, and the patient died on day 29 from the admission to the hospital. During the clinical course, we tried to manage the main clinical manifestations of this CAPS, and we tried to find possible steps to evaluate disease progression, such as platelet count during anticoagulation and pulse therapy, dysfunctions of the kidneys and heart, which were managed mainly with haemodialysis and we developed hypotheses with short outcomes to be evaluated at 7–14–21–28 days.

**Table 1.** Blood samples of patient with CAPS.

| Test | Normal Values | Day 1 Values | Day 3 Values | Day 5 Values | Day 7 Values | Day 9 Values | Day 15 Values | Day 20 Values | Day 24 Values | Day 29 Values |
|---|---|---|---|---|---|---|---|---|---|---|
| Haemoglobin (g/dL) | 12–15 | 9 | 8.2 | 8.2 | 8.5 | 8.9 | 9 | 8.5 | 8.4 | 8.2 |
| Platelets (mmcube) | 100–400 | 33 | 17 | 20 | 37 | 54 | 42 | 39 | 21 | 20 |
| WBC (mmcube) | 4–10 | 6.9 | 8.1 | 6.4 | 8.1 | 7.7 | 7.6 | 8.1 | 9.2 | 7.7 |
| Urea (mg/dL) | 21–45 | 133 | 87 | 105 | 110 | 115 | 109 | 86 | 179 | 152 |
| Creatinine (mg/dL) | 0.5–1.2 | 6.2 | 5.7 | 4.9 | 4.7 | 3.6 | 4.6 | 4.5 | 4.4 | 4.2 |
| Albumin (g/dL) | >2.8 | 2.9 | 2.9 | 3.0 | 2.8 | 2.9 | 3.1 | 2.9 | 2.9 | 2.9 |
| LDH (U/L) | 125–250 | 512 | 429 | 506 | 352 | 302 | 341 | 501 | 850 | 856 |
| d-dimer (ng/dL) | <500 | 2196 | 6196 | 3562 | 2156 | 1965 | 2003 | 1865 | 3154 | 1966 |
| C reactive protein (mg/dL) | <0.5 | 10.8 | 21.3 | 14.4 | 8.5 | 6.8 | 8.2 | 8.9 | 12 | 13 |
| Procalcitonin | | | | | | | | | | |
| Pro-BNP | <500 | 650 | 702 | 701 | 690 | 704 | 756 | 1980 | 5601 | >10,000 |
| Troponin (pg/dL) | 0–15 | 2564 | 7782 | 2156 | 1965 | 841 | 856 | 1950 | 1942 | 2112 |

**Table 2.** Thrombophilic screening of patient with CAPS.

| Test | Values | Normal Values |
|---|---|---|
| Protein C (%) | 91 | 80–120 |
| Protein S (%) | 93 | 80–120 |
| AT III (%) | 88 | 80–120 |
| Homocysteine (micromol/L) | 26 | <15 |
| Prothrombin time (INR) | 0.9 | 0.8–1.2 |
| Activated thromboplastin time (seconds) | 39 | 25–35 |
| Fibrinogen (mg/dL) | 396 | 200–400 |
| d-dimer (ng/mL) | 2196 | <500 |
| LAC (ratio) | 1.4 | <1.2 |
| Anticardiolipin IgM (U/MPL) | 12 | <10 |
| Anticardiolipin IgG (U/GPL) | 21 | <10 |
| Antibeta2 GP1 IgM (UI/mL) | 12 | <10 |
| Antibeta2 GP1 IgG (UI/mL) | 33 | <10 |
| ADAMTS-13 (%) | 37 | 35–150 |

**Table 3.** Thrombotic manifestation of reported patient and ongoing antithrombotic treatment.

| Type of Thrombosis | Ongoing Antithrombotic Treatment at Time of Diagnosis |
|---|---|
| Proximal right deep venous thrombosis | None |
| Renal vein thrombosis | None |
| Proximal left deep venous thrombosis | None |
| Arterial left anterior tibial thrombosis | Enoxaparin 40 mg daily |
| Right internal jugular vein thrombosis | Enoxaparin 80 mg daily |

## 3. Discussion

The clinical approach to CAPS is often difficult because the clinical scenario is frequently underhand. The objectives of a treatment for these complex diseases could be summarized as best anticoagulation to prevent recurrent thrombosis, best immunosuppressive treatment in order to reduce production of antiphospholipid antibodies, and treatment of damaged organs in order to reduce secondary diseases.

Yet, after diagnostic tools, a therapeutic approach to recurrent thrombotic events is difficult, in particular if kidney failure is present. Management of anticoagulants in CAPS as far as APS may be difficult because direct oral anticoagulants (DOACs) are not the first choice in APS, and treatments with warfarin frequently may offer off-label results with serial INR measurements [8,9]. In a similar way CAPS, as far as APS appear with severe thrombocytopenia and oral anticoagulation although needful may show complications as bleedings [10,11]. Furthermore, the management of all type of anticoagulants in patients with kidney failure with or without haemodialysis needs a particular attention with daily doses of low molecular weight heparins as far as warfarin as far as fondaparinux while DOACs are not suggested by guidelines [12]. In the case we described, in fact, when platelets' count increased, the best anticoagulant treatment was warfarin, as suggested by guidelines. Thereafter, further recurrent thrombosis disappeared. Furthermore, the differential diagnosis with other thrombotic microangiopathies made with ADAMS T13 testing helped us evaluate the best anticoagulant treatment because of the use of haemodialysis [13].

On the other hand, the management of kidney failure was also very hard. During the evaluation of the better antithrombotic treatment, kidney damage took relevance. Daily haemodialysis has been performed, but because the systemic blood pressure was unstable, the haemodialytic effect was not always effective. For this reason, the patient, day after day, made progress toward anasarca until heart failure.

This clinical complication was associated with a relapse of systemic inflammation. The short-term response to pulse therapy with steroids and IVIG was not followed by the similar clinical response to daily high doses of metilprednisolone and rituximab [14], so markers of inflammation showed a trend to increase while platelets decreased. On the other hand, someone may speculate that the recent COVID-19 infection may have a more relevant role in the occurrence of first thrombotic events as far as the trigger of APS [15]. During the pandemic, in fact, more than 30% of patients with COVID-19 showed the presence of antiphospholipid antibodies, and a part of them also developed APS without international diagnostic criteria [14], yet the occurrence of CAPS during COVID-19 infection has been rarely reported [15,16].

This clinical trend underlines one of the most serious troubles in clinical management of CAPS, which is related to the timing of follow-up and the damage of each organ caused by the syndrome, as reported in other clinical series. In other clinical experiences, a high rate of mortality occurred during the acute phase of the disease; furthermore, a relapse of APS signs and symptoms was detected in several patients with damage reported in similar organs as the kidney and heart [17]. In the case we reported, in fact, clinical improvements regarding antithrombotic treatments and platelet count were temporarily associated with

improvements in kidney failure and systemic inflammation, according to suggestions found in the CAPS registry [18], but after pulse therapy, these effects on kidney failure and systemic inflammation ceased. For these reasons, we found too many difficulties to join the follow-up of thrombotic manifestation and anticoagulant treatments with the follow-up of damaged organs such as the kidney and the heart. Of course, because there are no guidelines on this type for specific follow-up (i.e., to monitor multiorgan dysfunctions), a specific team should organize follow-up to evaluate different phases of treatments (i.e., acute treatment, sub-acute treatment, and long-term treatment) for each damaged organ in patients with CAPS. Therefore, one of the next steps in clinical series, clinical registries, and trials should be to establish periodical follow-up for each phase of the disease, which may include evaluation of each organ damaged during CAPS independently from the treatment of thrombosis.

**Author Contributions:** M.D. (Maurizio Dorato), M.L., M.D. (Maria D'Antò) and V.L. managed the case, P.D.M. wrote the article, G.S. reviewed proofs. All authors have read and agreed to the published version of the manuscript.

**Funding:** This research received no external funding.

**Institutional Review Board Statement:** The study was conducted in accordance with the Declaration of Helsinki, and approved by the Institutional Review Board of AFO Medicina.

**Informed Consent Statement:** Informed consent was obtained from all subjects involved in the study.

**Data Availability Statement:** The original contributions presented in the study are included in the article, further inquiries can be directed to the corresponding author/s.

**Conflicts of Interest:** The authors declare no conflict of interest.

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
