# Peer review of "Fulminant Recurrent Thrombosis in a Patient with Catastrophic Antiphospholipid Syndrome and Its Thirty-Day Outcome"

_2673-5601, doi:10.3390/immuno4010008_

Round 1

Reviewer 1 Report

Comments and Suggestions for Authors

Micco et al.'s 2024 manuscript details a case study involving a patient with Catastrophic Antiphospholipid Syndrome (CAPS), a rare clinical manifestation of Antiphospholipid Syndrome (APS) characterized by life-threatening complications arising from simultaneous thrombosis affecting both small and large vessels. The authors outline a series of clinical steps taken to manage this complex scenario and emphasize the importance of follow-up.

Within this case study, various diagnostic and treatment steps were undertaken. The authors are encouraged to elucidate:

The rationale behind choosing specific parameters for blood analysis and CT scans in the diagnostic process.

·         The criteria influencing their parameter selection.

·         what they expected what was the outcome and what they measure in reality

·         A review of existing literature addressing similar situations.

Additionally, the swift implementation of diverse medications for treatment warrants exploration:

·         The reasoning behind selecting particular treatments.

·         The expected outcomes of the chosen therapeutic interventions.

·         A review of published works relevant to the chosen treatment strategies.

The authors should consider presenting this manuscript at conferences involving experts from diverse fields related to the study. Post-discussion, incorporating commentaries into the manuscript could enhance its value, fostering advancements in the diagnosis and treatment of CAPS for future research and clinical practice.

Author Response

see attach

Reviewer 2 Report

Comments and Suggestions for Authors

It is difficult to attribute the multiple thrombotic events to APS since 1) the aPL titers are below the cut-off considered positive for a diagnosis of APS (based on both old Sapporo criteria and new ACR/EULAR criteria), 2) COVID-19 leads to a hypercoaguable state. Also, it should be stated that COVID-19 can lead to production of aPL serologies and it is unclear if these aPL cause a hypercoagable state (studies suggest they are not clinically relevant in COVID-19 thrombosis).

When did this patient present? The COVID-19 strain from 2020 tended to cause such fulminant thrombosis that mimicked CAPS. It is less common to see such severe presentations from COVID these days. Including the month and year of presentation in the report is critical. 

Were markers of thrombotic microangiopathy checked? With the anemia and thrombocytopenia, the reader wants to know more about the underlying cause. It is likely that this patient had microangiopathic hemolytic anemia and thrombocytopenia with multi-organ involvement.

Were other interventions such as plasma exchange considered?

This case is interesting from a "real-life challenging case" perspective, so if it is reframed as "Fulminant recurrent thrombosis in a COVID-19 patient", that may be a more accurate representation.

Comments on the Quality of English Language

English language needs to be improved.

Author Response

see attach
